# TGF-β Signaling in Progression of Oral Cancer

**DOI:** 10.3390/ijms241210263

**Published:** 2023-06-17

**Authors:** Yuanyuan Guo, Tiansong Xu, Yujuan Chai, Feng Chen

**Affiliations:** 1Department of Biomedical Engineering, Shenzhen University Medicine School, Shenzhen University, Shenzhen 518060, China; 248gyy@163.com; 2Key Laboratory of Optoelectronic Devices and Systems, College of Physics and Optoelectronic Engineering, Shenzhen University, Shenzhen 518060, China; 3Central Laboratory, Peking University School of Stomatology, Beijing 100081, China; willmaxu@163.com

**Keywords:** TGF-β signaling pathway, oral cancer, mechanism, therapeutics, oral squamous cell carcinomas, salivary adenoid cystic carcinoma, keratocystic odontogenic tumors

## Abstract

Oral cancer is a common malignancy worldwide, accounting for 1.9% to 3.5% of all malignant tumors. Transforming growth factor β (TGF-β), as one of the most important cytokines, is found to play complex and crucial roles in oral cancers. It may act in a pro-tumorigenic and tumor-suppressive manner; activities of the former include cell cycle progression inhibition, tumor microenvironment preparation, apoptosis promotion, stimulation of cancer cell invasion and metastasis, and suppression of immune surveillance. However, the triggering mechanisms of these distinct actions remain unclear. This review summarizes the molecular mechanisms of TGF-β signal transduction, focusing on oral squamous cell and salivary adenoid systemic carcinomas as well as keratocystic odontogenic tumors. Both the supporting and contrary evidence of the roles of TGF-β is discussed. Importantly, the TGF-β pathway has been the target of new drugs developed in the past decade, some having demonstrated promising therapeutic effects in clinical trials. Therefore, the achievements of TGF-β pathway-based therapeutics and their challenges are also assessed. The summarization and discussion of the updated knowledge of TGF-β signaling pathways will provide insight into the design of new strategies for oral cancer treatment, leading to an improvement in oral cancer outcomes.

## 1. Introduction

Oral cancer, with over 300,000 diagnoses worldwide annually and 145,000 deaths, is the sixth most common cancer. This major subtype of head and neck cancer affects 1.6 million people worldwide [1,2], but the five-year survival rate has only improved from 54% to 66% thanks to advanced treatment in the past two decades [3]. Oral cancer is most common in patients aged 45 to 64, comprising approximately 50% of all cases, with incidence increasing with age [4,5,6]. Over half of patients are diagnosed at advanced stages [7]. Nowadays, the primary treatments for oral cancer remain surgery resection, radio/chemotherapy, or combination treatment. Various factors should be taken into consideration when choosing treatment strategies, such as the tumor location, tumor size, general health conditions of the patients, and the accessibility of different methods to the doctors [8]. Given the high recurrence and metastatic rates in patients with advanced cancer, new therapeutic strategies must be developed to improve treatments. In some cases of oral cancers and tongue cancers, traditional treatments are difficult to implement, because the lesion is full of blood vessels, there is high risk of metastasis, and the patients may suffer from large surgical trauma and severe psychological stress. Consequently, exploring the molecular mechanisms at play is crucially important.

Transforming growth factor β (TGF-β), a cytokine, regulates many cellular processes, including cell growth, differentiation, development, and migration. Substantial evidence indicates that the TGF-β signaling pathway has two distinct roles: pro-tumorigenic and tumor-suppressive [9,10]. On the one hand, depending on cellular context, TGF-β may exert a tumor-promoting effect. In cells of late-stage cancer, its activation can promote tumorigenesis that affects cell motility, immune system evasion, metastasis, and neoangiogenesis. On the other hand, the TGF-β pathway exhibits tumor-suppressing functions in healthy cells as well as in early stage tumor cells; it inhibits epithelial cell-cycle progression and promotes apoptosis [9,10,11,12,13,14,15]. The dual functionalities and complex outcomes of TGF-β signaling render the therapies employing anti–TGF-β agents challenging; medication must be provided cautiously, and patient selection based on suitable indications is difficult.

Deregulation of TGF-β signaling contributes to oral cancer’s pathophysiology [16,17]. TGF-β upregulation has been detected in specimens collected from oral cancer patients with metastases from the mouth to bones [18]. TGF-β stimulates oral cancer cell migration [19,20], which implies TGF-β’s important role in cancer progression. Here, we review and summarize recent studies of TGF-β signaling regulation in oral cancers. The possible TGF-β signaling pathway therapeutic targets were addressed, with the discussion of current achievements and challenges. This review will provide insight into future applications of TGF-β-based medications.

## 2. Search Strategy

A comprehensive search of the literature was performed using the following search engines: PubMed, Web of Science, Embase, and Google Scholar. We focused on articles on oral squamous cell carcinomas, salivary adenoid cystic carcinomas, and keratocystic odontogenic tumors. The keywords used for searching were: ‘(TGF-β) AND (oral squamous cell carcinomas)’, ‘(TGF-β) AND (salivary adenoid cystic carcinomas)’, ‘(TGF-β) and (keratocystic odontogenic tumors)’, ‘(TGF-β) and (oral cancer therapy)’ and ‘(TGF-β) and (oral cancer)’. After the removal of the duplicates, the reviews, comments, non-English articles, and those without full text, there were 127 research papers left. The quality of these studies was evaluated, and studies with insufficient experimental evidence or not closely related to the scope of this review were excluded. Finally, a total of 59 studies were selected for this review (Figure 1).

## 3. TGF-β Signaling Pathways

The TGF-β superfamily of cytokines in humans consists of over 30 factors, including the TGF-βs (TGF-β1, -2, and -3); anti-Müllerian hormone (aka Müllerian inhibitory factor); activins; inhibins; nodal agents; myostatin; bone morphogenetic proteins (BMPs); and sundry differentiation and growth factors, which are found in all multicellular organisms [9]. TGF-β ligands are the primary TGF-β signaling mediators. Inactive TGF-βs are released as homodimeric polypeptides that interact with latency-associated peptide (LAP) and latent TGF-β-binding protein, forming a complex known as the large latent complex (LLC) that helps promote extracellular sequestration. LAP proteolysis and release of LLC from the extracellular matrix (ECM) requires TGF-β activation to free active TGF-β, which then binds to certain receptors [21]. Matrix metalloproteinase (MMP)-2 and MMP-9 work to cleave latent TGF-β and ECM protein. Thrombospondin 1 activates latent TGF-β [22]. Alternatively, via a conformational change, the αvβ6 integrin is capable of regulating TGF-β activity; it does so by binding LAP’s RGD motif, which in turn induces the latent complex to release mature TGF-β [23,24].

Members of the TGF-β superfamily bind to heteromeric receptor complexes containing type I (TβRI) and type II (TβRII) receptors with serine/threonine-kinase activities (Figure 2). Auxiliary coreceptors (also termed type III receptors, TβRIII) aid TGF-β superfamily members’ approaches to signaling receptors [25]. Activated TGF-β ligands interact with TβRII to form a heterotetrameric, an active receptor complex that recruits and phosphorylates TβRI (aka activin receptor-like kinase anaplastic lymphoma kinase [ALK] 5) at specific residues of serine and threonine. This complex, acting as a functional receptor, thus regulates downstream SMAD-dependent canonical pathway activation as well as SMAD-independent or noncanonical pathway activation [10,26].

The SMADs (mothers against decapentaplegic homolog) comprise proteins that are grouped together because they transduce extracellular signals directly to the nucleus. Mammalian cells contain eight different SMADs of three categories: (1) receptor-regulated SMADs (R-SMADs), which include SMAD1, SMAD2, SMAD3, SMAD5, and SMAD8/9; (2) the common mediator SMAD (Co-SMAD) SMAD4; and (3) inhibitory SMADs, including SMAD6 and SMAD7 (Figure 2). The major function of the inhibitory SMAD is to downregulate receptor-mediated R-SMAD phosphorylation. This results in the prevention of complex formation with Co-SMAD. In the SMAD-dependent pathway, phosphorylated TβRI recruits and phosphorylates R-SMADs (SMAD2 and SMAD3). SMAD2 and SMAD3, in association with SMAD4, then modulate transcription. After activation, the SMAD complex is translocated to the nucleus, where it binds specific DNA domains known as SMAD-binding elements. Upon binding, R-SMAD–Co-SMAD complexes interact with other DNA-binding transcription factors to transactivate or repress target genes. In addition to such canonical signaling, the activated TGF-β receptor complex transmits signals via other SMAD-independent pathways, including the mitogen-activated protein kinase (MAPK) pathway, the c-Jun N-terminal kinase (JNK)/p38 pathway, and the phosphatidylinositol-3 kinase (PI3K)/protein kinase B (AKT) pathway; it transmits signals by tumor necrosis factor (TNF) receptor-associated factor 4 (TRAF4), TRAF6, and Rho GTPases [26,27]. Several signaling pathways are implicated in TGF-β signaling, including Notch, Wnt, PI3K/AKT, interferon, TNF, and renin-angiotensin system (aka RAS) pathways [9,28].

## 4. Role of TGF-β in Oral Squamous Cell Carcinomas

Oral squamous cell carcinomas (OSCCs) occurring in the mucus membranes of the mouth and/or oral cavity account for >90% of all oral malignancies [29]. OSCCs can develop from oral precancerous lesions such as leucoplakia and erythroplakia [30]. In most patients, when OSCC is diagnosed, the disease is advanced (stages III and IV) [31]. In OSCC, the role of TGF-β signaling is complex because TGF-β evidences both anti- and protumorigenic activities. Early in carcinogenesis, TGF-β acts as a potent tumor suppressor. Yet, during advanced disease stages, TGF-β may promote cancer cell growth and/or metastasis growth, thus advancing tumor progression [32].

In early stage OSCC, TGF-β exhibits potent inhibitory growth effects; it promotes apoptosis, reconstitutes the tumor microenvironment, and suppresses cell cycle progression via G_1_ arrest [33]. The mechanisms of such TGF-β-mediated arrest differ depending on cell type and cell differentiation stage. However, in most squamous carcinoma cells, TGF-β suppresses cell-cycle proliferation via G_1_ arrest, including the inhibition of *MYC* and cyclin-dependent kinase (CDK) 4 (CDK4), and repression of cyclin activities and induction of the CDK inhibitors p21 (aka CIP1) and p15 (aka INK4B) [34]. TGF-β represses *MYC* transcription, which encodes a transcriptional activator of genes required for cell growth and proliferation. The SMAD2/SMAD3/SMAD4 complex acts together with Sp1 of the *p15* promoter to induce *p15* gene expression in response to TGF-β1 [34,35]. The p15 protein suppresses the kinase activities of the cyclin D–CDK4/6 complex and inhibits phosphorylation of intracellular target retinoblastoma protein, which is necessary for G_1_/S progression. TGF-β1-induced p21 plays a critical role in the inhibition of cyclin-CDK kinase activities, increasing p27 binding to cyclin E-CDK2 and inhibiting the expression of cyclin D1, CDK4, and CDK6 to mediate cell cycle arrest in G_1_ [36,37]. Wang et al. found that TGF-β1 rapidly increased p15 and p21 expression in both high- and low-metastatic OSCC cell lines. OSCC growth arrest might be thus induced by TGF-β1 via multiple pathways [38].

Studies on TGF-β-mediated tumor suppression revealed that loss of TGF-β signaling components is associated with carcinoma progression [10,39,40]. In OSCC patients, TβRII expression is significantly lower in metastases than in the primary tumors, where TβRII expression is downregulated compared to the normal oral epithelium [16]. Fukai et al. found that decreased TβRII expression correlates with high cell migration, high invasiveness, and poor prognosis of OSCC [36]. In OSCC patients, it is common that TβRII and TβRIII lack expression in oral epithelium and stroma; however, the TβRI expression levels in OSCC do not change significantly compared to normal tissue [16,41]. Mutation of genes encoding *SMAD* transcription factors (*SMAD2*, *SMAD4*) revokes TGF-β signaling’s tumor-suppressive effects and triggers OSCC progression [42].

During advanced carcinogenesis stages, TGF-β promotes OSCC invasion and metastasis, acting by many pathways, including suppression of immune surveillance, induction of the epithelial-mesenchymal transition (EMT), and modulation of the tumor microenvironment to promote carcinoma-cell proliferation. The EMT results in cell–cell adhesion loss, thus increasing cell mobility and promoting metastasis. During EMT, cells lose E-cadherin as well as other components of epithelial cell junctions. Metastasis is accelerated by the EMT, which is associated with increased cell migration and invasion, and increased cell-substrate adhesion. The overexpression of TGF-β ligands is common in OSCC as it promotes EMT and cell invasion in vitro [43,44]. Of all the inducers of EMT-activating transcription factors, including Slug, Snail, Twist, and Zeb1, TGF-β is among the best known [45]. Sun et al. found that TGF-β1 regulates MMP-9 expression to enhance the EMT of OSCC cells by inducing the expression of Snail/Ets-1 [43]. TGF-β1 induces MMP-9 expression via the SMAD/myosin light chain kinase pathway and upregulates Slug by acting on extracellular-regulated protein kinases (ERK) 1/2 to promote OSCC invasion and metastasis [46,47].

Immune surveillance is crucial in carcinogenesis. TGF-β and the cytokine interleukin (IL)-10 are typical anti-inflammatory immunosuppressive cytokines that promote immune system evasion by neoplastic cells [48,49]. In OSCC, TGF-β and IL-10 upregulation, combined with the downregulation of natural killer (NK) cell-activating cytokines such as IL-2 and NK receptors, modulate the immune escape of NK cells [50,51]. TGF-β downregulates IL-2 expression and thus suppresses T-cell proliferation, activation, and differentiation [52]. In a study of head and neck squamous cell carcinoma using an orthotopic murine model, Dasgupta et al. found that TGF-β1 reduces the expression of the NK-activating receptor NKG2D on the NK cell surface and inhibits NK cell functions [53]. T_H_17 cells produce IL-17 and participate in various inflammatory reactions. T_H_17 cell numbers and upregulated IL-17 levels correlate with the extent of OSCC malignancy. TGF-β plays a proinflammatory role via its effects on cells of the innate immune system as well as on T_H_17 cells [54,55].

In addition, TGF-β induces macrophages to differentiate into the M2 phenotype in the OSCC tumor microenvironment. A predominance of M2 macrophages triggers local immunosuppression and more lymph node involvement and reduces survival [56]. In tumor microenvironments, cancer-associated fibroblasts (CAFs) are central to immunosuppression. In an ex vivo model of OSCC, TGF-β, IL-10, and arginase-1 caused CAFs to induce CD14 myeloid cells to develop an M2-like phenotype, ultimately suppressing T-cell proliferation and further promoting tumorigenesis [57].

## 5. Role of TGF-β in Salivary Adenoid Cystic Carcinoma

One of the most common malignant tumors of the salivary glands is salivary adenoid cystic carcinoma (SACC), which is associated with perineural and perivascular invasion, distant metastasis, and poor prognosis. SACC exhibits certain specific characteristics, including multiple local recurrences, indolent growth, and high metastasis incidence [58,59,60]. Treatment failure, as in most solid tumors, is generally the result of distant metastasis at an early disease stage [61]. TGF-β both suppresses and promotes SACC activities. However, further systematic research on TGF-β signaling in SACC is needed.

TβRIII expression is markedly repressed in SACC patients. Xu et al. demonstrated that transient TβRIII overexpression induces SACC m apoptosis and G_2_/M arrest that inhibit cell growth and migration; TβRIII, scaffolding protein arrestin 2 (β-arrestin 2), and IκBα form a complex that is an important negative regulator of NF-κB signaling and promoting the TβRIII-mediated inhibition of SACC migration and progression [62]. A member of the TGF-β superfamily, Runt-related transcription factor 3 (RUNX3) acts as a tumor suppressor gene and induces apoptosis. After CpG island hypermethylation of the promoter region in various carcinomas such as SACC, RUNX3 expression is decreased or even abolished [63]. Poor prognosis, a tumor located in a salivary gland, and SACC recurrence all correlate highly with *RUNX3* downregulation associated with DNA hypermethylation and protein mislocalization. RUNX3 mislocalization may be a novel mechanism of *RUNX3* gene inactivation that impairs TGF-β signaling [63]. TGF-β-stimulated clone 22 (TSC-22) induces differentiation, thus negatively regulating salivary-gland cancer cell growth. In SACC patients, TSC-22 was hyper expressed in the inner ductal cells of tubular structures. Nuclear translocation of TSC-22 is essential for the induction of salivary-gland cancer cell apoptosis [64]. Dong et al. found that the upregulation of TGF-β1 and phospho-SMAD2, combined with the downregulation of membrane β-catenin, is associated with pulmonary metastasis in SACC patients; they also found that TGF-β1 promotes the invasion and migration of SACC cell lines via the SMAD pathway, inducing the EMT and changing the morphology of SACC cells [65]. The redox protein thioredoxin 1 is critical in terms of cellular redox homeostasis regulation and the promotion of antiapoptotic functions. One study found that TGF-β-induced EMT is mediated by thioredoxin 1, and the EMT further promotes SACC metastasis by stabilizing Slug and Snail, two transcription factors associated with PI3K/AKT/glycogen synthase kinase 3β signaling [66]. TRAF6 mediates SACC progression by activating SMAD2/SMAD3/p38/JNK/ERK signaling cascades [67].

## 6. Role of TGF-β in Keratocystic Odontogenic Tumors

Keratocystic odontogenic tumors (KCOTs) are benign odontogenic cystic epithelial neoplasms that comprise approximately 11% of all oral cysts [68]. KCOTs are single-cell or multicell, rapidly growing, and intraosseous tumors associated with nevoid basal cell carcinoma syndrome (NBCCS). Human KCOTs arise in two ways: sporadically, or as components of this syndrome. In contrast to other maxillary cysts, such as epidermoid and radicular cysts, KCOTs have high recurrence rates. They may transform malignantly and exhibit a characteristic lining of parakeratinized, stratified squamous epithelium [69,70,71,72].

Some human KCOTs obtain *PTCH1* mutations that may also occur in tumors associated with NBCCS as well as in the sporadic forms of retinoblastoma, basal cell carcinoma, and medulloblastoma [73,74]. *PTCH1* mutations may derepress Smoothened (Smo) and constitutive Sonic Hedgehog (Shh) signal activities [74]. When Shh binds Patched that is encoded by *PTCH1*, then the inhibition of Smo by Patched ceases. Upon the activation of the Hedgehog pathway, the downstream transcription factor Gli will enter the nucleus to initiate the transcription of downstream effector genes, including *PTCH1* per se as well as Wnt and TGF-β/BMP family genes [75]. The aberrant inactivation of *PTCH1* triggers constitutive activation of the Hedgehog pathway, causing a series of developmental abnormalities and tumorigenesis [74,76]. It is unknown whether other signal transduction events occur during KCOT initiation and progression. It is believed that tooth development is closely related to dental epithelial status. However, the pathology that bridges tooth root development to KCOT tumorigenesis remains under investigation.

Previous studies have suggested that TGF-β might play roles in various lesions, including odontogenic cysts, by regulating epithelial cell differentiation [77]. TGF-β may be involved in local invasiveness [78], biological behaviors [79], and the proliferation and/or differentiation of various odontogenic cysts and tumors. It has been suggested that TGF-β expression may differ among these lesions, thus affecting epithelial-mesenchymal interaction as well as extracellular matrix formation and growth [77,80,81]. The expression of TGF-β1 and its receptors correlated negatively with the differentiation of OSCC [82] and other cancers [83,84], but differentiation was further enhanced in terminally differentiated mucosal keratinocytes [85]. Santos et al. found that the level of immunoreactive TGF-β1 in connective tissue is higher in KCOTs than in other cysts [86].

TGF-β acts in all stages of tooth development by the regulation (paracrine and autocrine) of epithelial-mesenchymal interactions, regulation of odontoblast proliferation, maintenance of homeostasis of odontogenic epithelium, and facilitation of terminal differentiation of ameloblasts and odontoblasts [79,87,88,89]. Gao et al. suggested that *SMAD4* serves as a tumor suppressor gene of the TGF-β/BMP pathway, plays important roles in epithelial–mesenchymal interactions that control root development, and suppresses multiple KCOTs in the jaw [90]. A reduction in SMAD4 expression changes the fate of odontoblasts, dramatically reactivates Hedgehog signaling within the hyperplastic epithelium, and alters the fates of the Hertwig epithelial root sheath and the epithelial rests of Malassez, thus triggering the formation of multiple jaw odontogenic cysts [81]. Yamada et al. reported that TGF-β and IL-1α in KCOT fluid act together to induce receptor activator of NF-κB ligand (aka RANKL) expression in stromal fibroblasts and to promote osteoclast formation that might mediate jawbone resorption [91]. Zhong et al. found that the presence of both Slug and MMP-9 in KCOT samples indicates that the EMT profoundly affects local invasion by KCOTs [92].

## 7. TGF-β as a Therapeutic Biomarker

The TGF-β signaling pathway plays both tumor-promoting and tumor-suppressing roles in cancer (Figure 3). TGF-β potently inhibits epithelial cell growth via the induction of various proteins that inhibit the cell cycle. Thus, during early oral carcinogenesis, TGF-β strongly suppresses tumors, maintaining tissue homeostasis and inhibiting tumor progression [93,94,95]. Patient stratification based on gene expression has revealed that the highest TGF-β pathway activity is associated with the worst prognosis. TGF-β is notably overexpressed in aggressive and invasive cancers; it triggers malignant effects in established cancers by induction of the EMT and metastasis [96]. Thus, TGF-β promotes tumor invasion, growth, and metastasis. The dual TGF-β cancer phenotypes are in part dependent on the disease progression stage (Figure 3).

TGF-β may be a potential cancer-drug target when TGF-β is overexpressed and thus promotes tumor growth. However, the usefulness of such drugs is compromised by the dual roles that TGF-β signaling plays in cancer. TGF-β promotes fibrosis, angiogenesis, and metastasis as well as suppressing host immune responses, thus enhancing the tumor microenvironment at all carcinogenetic stages. The TGF-β pathway is therefore a candidate for a microenvironment-targeted cancer treatment.

Aberrant TGF-β signaling contributes to the pathogenesis of many diseases. Several TGF-β pathway inhibitors have been preclinically investigated, with some currently under clinical evaluation. For such applications, the downregulation of TGF-β expression and the inhibition of TGF-β-receptor kinase activation is used to suppress TGF-β signaling [97]. Various methods have been used to antagonize TGF-β actions, such as the usage of antisense oligonucleotides that target specific TGF-β signaling components. These treatments are either provided intravenously or engineered into immune cells to prevent TGF-β synthesis [98,99]. Monoclonal antibodies of the anti–TGF-β receptor prevent ligand–receptor interactions [100], and small-molecule inhibitors of TGF-β receptor kinase prevent signal transduction. Engineered mutant TGF-β ligands [98], recombinant soluble β-glycan [101,102], decorin [103], and a soluble TβRII:Fc fusion protein [104,105,106] all inhibit TGF-β activity or signaling.

TGF-β inhibitors suppress tumor development and reduce fibrosis in tumor microenvironments. A dense fibrotic stroma is a characteristic of several tumors, including pancreatic ductal adenocarcinomas. TGF-β regulates the functions of stromal cells in fibrotic tissues and tumor microenvironments by driving tissue angiogenesis, inducing immune cell recruitment, and promoting fibroblast differentiation into myofibroblasts [107]. TGF-β mediates myofibroblast differentiation and activates resident/infiltrating fibroblasts. CAFs and stellate cells are activated by mitogenic and fibrogenic stimulants, including TGF-β, platelet-derived growth factor, and Shh released by cancer cells. CAFs and stellate cells, when activated, emit factors that promote tumor growth, invasion, and metastasis [108]. In the Panc-1 orthotopic pancreatic ductal adenocarcinoma model, TGF-βRI inhibition by SD-208 significantly reduced the biological activities of fibroblasts and suppressed tumor growth [109]. Medicherla et al. found that the blocking of TGF-β signaling with galunisertib or LY2109761 reduces tumor stromal content and inhibits the synthesis of connective tissue growth factor by downregulating fibroblastic-cell proliferation in hepatocellular carcinomas [110]. Pirfenidone (5-methyl-1-phenylpyridin-2-one, brand name Esbriet), an orally available pyridine derivative, was approved in 2014 by the US Food and Drug Administration to treat idiopathic pulmonary fibrosis (Table 1). Although the precise biological target of pirfenidone remains unclear, pirfenidone inhibits TGF-β production, thereby reducing collagen synthesis and fibroblast proliferation and delaying idiopathic pulmonary fibrosis development [111]. Fluorofenidone, which is a structural analog of pirfenidone, also inhibits TGF-β expression. Fluorofenidone suppresses the TGF-β-induced or -associated EMT and attenuates myofibroblast differentiation [112,113]. Malignant transformation to OSCC develops in approximately 7% of patients with oral submucous fibrosis. In such patients, small-molecule TGF-β inhibitors that may treat oral cancers are an alternative to surgery.

The inhibition of TGF-β signaling is an emerging cancer therapy strategy. Most small-molecule inhibitors of such signaling specifically target the type I receptor of TGF-β to inhibit the phosphorylation of SMAD2 and SMAD3 with the maintenance of at least some noncanonical responses. TβRI kinase inhibitors affect the immune response, invasion/metastasis, and extracellular matrix production. SB-431542 inhibited the kinase activities of TβRI, ALK4, and ALK7; inhibited the phosphorylation of SMAD2/SMAD3; and reduced cell proliferation and migration [114,115,116]. TEW-7197 (vactosertib) is a competitive inhibitor of ATP that inhibits kinase activity by binding the hydrophobic ATP pocket of the TβRI intracellular domain. In a mouse model, TEW-7197 effectively inhibited melanoma development and lymph-node metastasis. In addition to blocking the phosphorylation of SMAD2/SMAD3, TEW-7197 enhanced the antimelanoma cytotoxic T-lymphocyte immune response by increasing the proliferation of CD8^+^ T lymphocytes in mice with melanoma by ubiquitin-mediated SMAD4 degradation [117]. SMAD2 phosphorylation induced by TGF-β1 was downregulated by LY2157299 in human HLE and HLF hepatocellular carcinoma cells. Additionally, LY2157299 significantly inhibited cancer-cell metastasis and proliferation [118].

Despite the accumulated evidence of TGF-β-targeting therapeutics, it should also be noticed that inhibitors of TGF-β receptor kinases sometimes have poor pharmacokinetics and pharmacodynamics. They target TGF-β receptors non-specifically, and inhibit related type I receptors for several other TGF-β-related proteins equally effectively [115], and may also inhibit other kinases, such as p38 MAPK [119]. The lack of specificity and poor pharmacokinetics of current TGF-β receptor kinase inhibitors lead to potential challenges with dosing strategies. Due to the aforementioned issues of the TGF-β-pathway-targeted therapeutics, it is critical to identify reliable biomarkers that predict response to TGF-β inhibitors. Predictive capacities may differ depending on the tumor microenvironment and overall blood biomarker spectrum of individual patients [120]. Zinc finger E-box-binding homeobox 1 (ZEB1) is a critical activator of the EMT associated with cancer. The direct targeting of ZEB1 by miR-101 induced EMT, and the downregulation of miR-101 promoted OSCC growth and metastasis [121]. MiR-455-5p suppresses UBE2B and thus contributes to oral tumorigenesis by binding specific promoter regions of SMAD3 [122]. αvβ6 regulates TGF-β activity by binding the RGD sequence of LAP, triggering cancer progression [123,124]. The blockade of αvβ6 did not affect tumor cell proliferation in vitro but inhibited tumor growth in vivo [125]. The microenvironment thus plays a role in such regulation; further work in this realm is necessary.

## 8. Discussion and Conclusions

In the tumor microenvironment, TGF-β is becoming an important and clinically actionable means of immune evasion [126]. Meanwhile, TGF-β signaling inhibition is also an emerging strategy for cancer therapy. There are three potential strategies: (1) inhibiting TGF-β signaling components or modulating factors; (2) blocking the interaction between TGF-β and other signaling pathways in cancer; and (3) normalizing tumor microenvironment homeostasis by downregulating stromal stimulation resulting from excess TGF-β produced by tumors and tumor-related tissues.

TGF-β-pathway inhibitors have been investigated in the preclinical setting, some of which are now in the clinical development phase [127]. However, unlike preclinical results, TGF-β blocking progress in clinical trials has always been difficult, and many trials have failed to replicate success in animal models. Due to the dual role of the TGF-β signaling pathway, the window of effective TGF-β targeting is therefore evidently small, which poses a clear challenge in selecting patients at the right time. The ideal evaluation method will enable the identification of individuals who will benefit from TGF-targeted therapy, as well as excluding patients in whom TGF-targeting will create limited or even detrimental effects. Alternatively, another potential reason for the relative lack of success in clinical trials is that immune-suppressive TGF-β signaling may be more nuanced than previously realized, and additional factors must be considered when designing anti-TGF-β therapies. Since the effects of TGF-β signaling are both localized and rapid [15], it is important to choose the dose and frequency of administration to support the sustained inhibition of the TGF-β receptors and prevent the reactivation of TGF-β signals in the tumor microenvironment. In addition, the identification of reliable and predictable biomarkers of response to TGF-β inhibitors is also a critical issue.

TGF-β signaling plays a key role in cancer progression and high TGF-β signaling activity has been linked to resistance to multiple anticancer therapies, including chemotherapy and molecularly targeted therapies [127]. Enhanced intratumoral TGF-β signaling is also a barrier in patients in response to immunotherapy. Given the role of TGF-β signaling in non-pathophysiological processes, it is not surprising that systemic inhibition of TGF-β signaling would accompany the onset of adverse events. However, current early clinical trials of TGF-β inhibitors show that adverse events are generally controllable, such that therapies respond significantly and persistently when combined with immune checkpoint inhibitors. In the future, the inhibition of TGF-β with dual-specific drugs or using drug interference with the composition of TGF-β signaling with restricted tissue expression are expected to lead to new therapies. Furthermore, combining TGF-β inhibitors with cytotoxic drugs, radiotherapy, adoptive T cell transfer therapy (such as CAR-T), immune checkpoint inhibitors (such as PD-1/PD-L1 antibodies), cancer vaccines, viral vectors, nanoparticles, or oncolytic viruses may provide additional opportunities.

## 9. Future Implications

Our understanding of oral cancer has been greatly improved by the developments in molecular biotechnology and human genetics. TGF-β signaling plays a complex role in oral cancer progression and development, as it can serve as a tumor promoter or suppressor, depending on the progression stage of oral cancer. However, much remains to be investigated in the future, particularly the cross-talk between the TGF-β pathway and other signaling pathways. Further elucidation of regulatory tumor-host interactions may identify more useful therapeutic targets and aid in the accurate timing of TGF-β-targeted therapy and clinical development of new TGF-β inhibitors.

TGF-β and other components of the TGF-β signaling pathway are also therapeutic antitumor candidates. Future work on the TGF-β signaling mechanisms that operate in oral cancer may reveal new therapeutic targets, such as small-molecule TGF-β inhibitors, that broaden the range of drug development and even prevent or cure oral cancer.

## Figures and Tables

**Figure 1 ijms-24-10263-f001:**
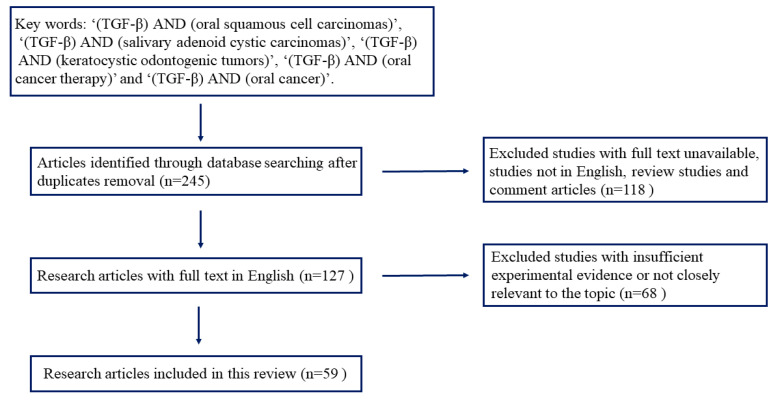
Flowchart of the search strategy and the literature selection process.

**Figure 2 ijms-24-10263-f002:**
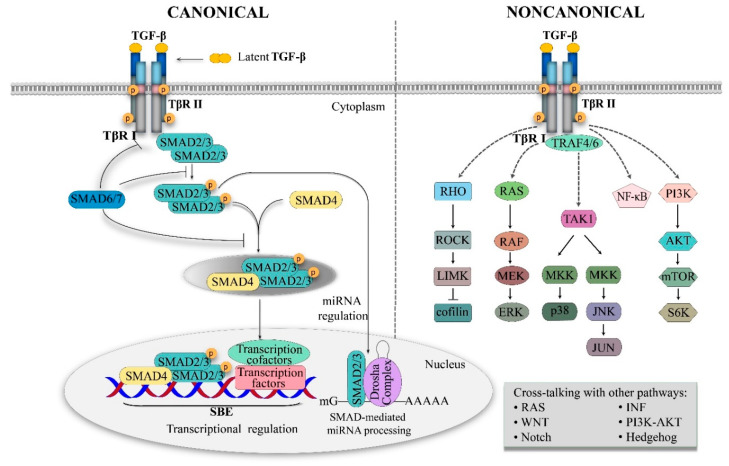
Overview of TGF-β signaling pathway. TGF-β signaling is transduced through 2 pathways: canonical (SMAD dependent) and noncanonical (SMAD independent). Signal transduction initiates by binding TGF-β to TβRII, which recruits and transphosphorylates TβRI. Either canonical or noncanonical signaling pathways can subsequently be activated by the TβRI–TβRII complex. After phosphorylation, TβRI becomes activated, then activates the canonical pathway through phosphorylation of R-SMADs (SMAD2 and SMAD3). R-SMADs form heteromeric complexes with Co-SMAD (SMAD4) and translocate to the nucleus, where they bind to site-specific recognition sequences (SBEs) and regulate the expression of TGF-β-signaling-pathway target genes. R-SMADs modulate the biogenesis of microRNA (miRNA) by promoting the processing of primary miRNA into precursor miRNA in the nucleus. mG, 5′ capping of messenger RNA (mRNA); AAAAA, 3′ polyadenylation of mRNA. In noncanonical pathways, TGF-β signaling activates SMAD-independent pathways such as the renin–angiotensin system (RAS)/ERK, PI3K/AKT, JNK and p38, NF-κB, and Rho. TGF-β signaling can be influenced by other pathways (e.g., RAS, Wnt, Hedgehog, interferon, TNF, Notch).

**Figure 3 ijms-24-10263-f003:**
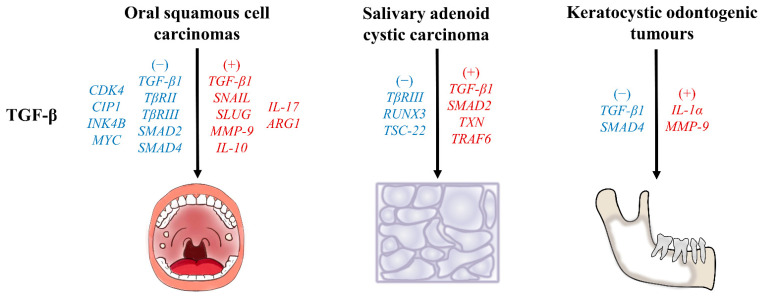
Relationship between genes related to TGF-β signaling pathway and pathogenesis of oral cancers. The diagram outlines the relationship between genes related to the TGF-β signaling pathway and the pathogenesis of oral cancers, including OSCC, salivary adenoid systemic carcinoma, and KCOTs. Genes indicated in red might promote disease progression; genes indicated in blue might suppress disease progression.

**Table 1 ijms-24-10263-t001:** Overview of clinical trials for TGF-β targeting agents.

Drug	Type	Targets	Disease Applications	Clinical Trial Identifier	Phase
Trabedersen (AP12009)	Antisense oligo	TGF-β2 ligand	Melanoma, pancreatic cancer, colorectal carcinoma	NCT00844064	Phase I
			Glioblastoma	NCT00431561	Phase I/IIB
			Glioblastoma	NCT00761280	Phase III
Belagen-pumatucel-L (Lucanix)	Antisense gene-modified allogeneic tumor cell vaccine	TGF-β2	Non-small cell lung cancer	NCT00676507	Phase III/IV
			Lung neoplasm, bronchogenic carcinoma	NCT01058785	Phase II
Fresolimumab (GC-1008)	Humanized antibody	TGF-β1, TGF-β2, TGF-β3	Renal cell carcinoma	NCT00923169	Phase I
			Renal cell carcinoma, melanoma	NCT00356460	Phase I
			Primary brain tumors	NCT01472731	Phase II
			Metastatic breast cancer	NCT01401062	Phase II
			Non-small cell lung carcinoma	NCT02581787	Phase I/II
			Pleural malignant mesothelioma	NCT01112293	Phase II
Galunisertib (LY2157299)	Small molecule	TβRI kinase	Advanced solid tumors, pancreatic cancer	NCT01373164	Phase I/II
			Hepatocellular carcinoma	NCT02240433	Phase I
			Advanced solid tumors	NCT01722825	Phase I
			Malignant glioma	NCT01220271	Phase I
			Hepatocellular carcinoma	NCT01246986	Phase II
			Advanced hepatocellular carcinoma	NCT02178358	Phase II
			Pancreatic neoplasms	NCT02154646	Phase I
			Metastatic pancreatic cancer	NCT02734160	Phase I
			Metastatic androgen receptor negative (AR-) triple negative breast cancer	NCT02672475	Phase I
			Recurrent malignant glioma	NCT01682187	Phase I
			Advanced solid tumors, non-small cell lung carcinoma, hepatocellular carcinoma	NCT02423343	Phase I/II
			Advanced solid tumors	NCT02304419	Phase I
Pirfenidone	Small molecule, not TGF-β specific		IPF, glomerulosclerosis and diabetic kidney disease, pathological skin scarring	Multiple trials	Phase III
TEW-7197	Small molecule	TβRI kinase	Advanced solid tumors	NCT02160106	Phase I
			Myelodysplastic syndromes	NCT03074006	Phase I

## Data Availability

Not applicable.

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
