# Peer review of "TGF-β Signaling in Progression of Oral Cancer"

_ijms, 2023, doi:10.3390/ijms241210263_

Round 1

Reviewer 1 Report

The manuscript entitled TGF-β signaling in progression of oral cancer, by Guo Y et al, addresses a very interesting topic, which is highly relevant for the research in the field, given the high incidence and mortality rate of oral cancer. The authors present the manuscript clearly and is well written and well structured, having included 2 figures and 1 table that make the informtion presented in the text easier to understand. In my opinion, this commprehensive review would be very interesting for the readership of this journal, therefore I recommend its publication in present form.

Author Response

 We sincerely appreciate your comments. We would also like to thank you for your kind support and appreciation of our study.

Point 1: The manuscript entitled TGF-β signaling in progression of oral cancer, by Guo Y et al, addresses a very interesting topic, which is highly relevant for the research in the field, given the high incidence and mortality rate of oral cancer. The authors present the manuscript clearly and is well written and well structured, having included 2 figures and 1 table that make the information presented in the text easier to understand. In my opinion, this comprehensive review would be very interesting for the readership of this journal, therefore I recommend its publication in present form.

Response 1: We sincerely appreciate your comments. We would also like to thank you for your kind support and appreciation of our study.

Reviewer 2 Report

It is an interesting review given the fact that the incidence of oral cancer is increasing globally. TGF- signaling has recently been shown to play an important role in patients receiving certain anti-cancer therapies. Chemotherapeutics can stimulate TGF- production and, as a result, increase TGF- signaling in the tumor microenvironment. Thus, TGF-signaling inhibition emerges as an urgent need in cancers resistant to available therapies. Various TGF-inhibitors have been used alone or in combination with current cancer therapies in experimental human clinical trials, with varying degrees of success.

I have the following comments:

1. The authors must include a section of "Methodology", and the authors must mention the inclusion and exclusion criteria employed for selection of literature for this review.

2. Mention the date between which the articles for this review were selected, and which databases were mined for literature survey (PubMed, Science direct, etc??)

3. There are some typing errors and grammatical mistakes. Please revise the manuscript by a native English speaker.

4. Include a section "Future Implications" before the references.

There are some typing errors and grammatical mistakes. Please revise the manuscript by a native English speaker.

Reviewer 3 Report

This review provides detailed insights into the role of the Transforming Growth Factor-beta (TGF-β) signaling pathway in the pathogenesis of various oral diseases, such as salivary adenoid cystic carcinoma (SACC), keratocystic odontogenic tumors (KCOTs), and oral squamous cell carcinoma (OSCC). The authors also discuss the potential of targeting TGF-β as a therapeutic intervention. However, some critical perspectives on the review should be taken into account and some areas could be improved:

1) While the review provides a lot of detailed information about the role of TGF-β in oral diseases, it seems to heavily lean towards the molecular mechanisms and may not be accessible to a broader audience without a deep background in molecular biology or genetics.

2) The review does not discuss any contrary evidence or studies that challenge the proposed roles of TGF-β. The inclusion of counter-arguments or contrasting viewpoints can provide a more balanced perspective.

3) Although the review discusses the potential of TGF-β as a therapeutic target, it does not provide a clear conclusion or suggest future research directions.

4) The review does not critically assess the quality or limitations of the cited studies, which is crucial for a more nuanced understanding of the topic.

5) Given the dual role of TGF-β in tumor suppression and promotion, there could be a more balanced discussion on this aspect, including potential pitfalls and challenges in targeting TGF-β as a therapeutic strategy.

Overall, the review provides a comprehensive overview of the role of TGF-β in oral diseases, with a focus on the molecular mechanisms involved. However, it could benefit from more balanced content, a clear conclusion, a critical evaluation of cited studies, and a discussion of potential challenges in TGF-β targeting for therapy.

Minor point:

Authors could have provided a broader perspective on TGF-β's role in various cancers at the start of the review. This would set the stage for readers by providing a more comprehensive understanding of TGF-β's multifaceted roles in cancer before diving into the specifics of oral diseases.

Here are a few references that discuss TGF-β's role in various cancers:

1) Massagué, J. (2008). 10.1016/j.cell.2008.07.001. This seminal review provides a comprehensive overview of TGF-β's roles in various types of cancer.

2) Pickup, M.. (2013). 10.1038/nrc3603. This review specifically addresses how TGF-β acts within the tumor microenvironment, which is relevant to many types of cancer.

3) Ikushima, H. (2010). 10.1038/nrc2853.

Here are some references that delve into the role of TGF-β in specific types of cancer that authors should consider:

1) Breast Cancer: Bierie, B. (2006). 10.1038/nrc1926. This review discusses the dual role of TGF-β in breast cancer, acting as a tumor suppressor early on and a tumor promoter later.

2) Colorectal Cancer: Calon, A., (2012). 10.1016/j.ccr.2012.08.013. This study illustrates the role of TGF-β in promoting metastasis in colorectal cancer.

3) Pancreatic Cancer: Principe, D. R. (2014). 10.1093/jnci/djt369. This review discusses the complex role of TGF-β in pancreatic cancer, a notoriously difficult-to-treat disease.

4) Lung Cancer: Giannos P. (2021). 10.3390/biology10111200. This article explores the role of TGF-β in lung cancer, particularly how its activation contributes to disease prognosis.

5) Liver Cancer: Giannelli, G. (2014). 10.1158/0008-5472.CAN-14-0243. This review discusses the potential for targeting TGF-β in the treatment of liver cancer.

Minor editing of English language required. Presence of a few syntax errors with colloquial statements.

Author Response

Response to Reviewer 3 Comments

Point 1: While the review provides a lot of detailed information about the role of TGF-β in oral diseases, it seems to heavily lean towards the molecular mechanisms and may not be accessible to a broader audience without a deep background in molecular biology or genetics.

Response 1: Thanks for your constructive suggestions. This review is aimed to the special topic of “TGF-β signaling in Human disease” under molecular pathology section, so we laid more efforts to elaborate the molecular mechanisms in depth to fit in the topic. Small molecule inhibitors of specific signaling pathways have been widely used as a routine treatment for tumors. When we treated the tumors, the signaling pathways were considered frequently. According to your comment and for a better accessibility to the broader audiences, especially the readers without a deep background in molecular biology or genetics, we rephrased the introduction section and added the content of introduction to current regular oral cancer treatments.

The corresponding paragraph was included as follows: Nowadays, the primary treatments for oral cancer remain to be surgery resection, radio/chemotherapy, or combination treatment. Various factors should be taken into consideration when choosing treatment strategies, such as the tumor location, tumor size, general health conditions of the patients, and the accessibility of different methods to the doctors [8]. Given the high recurrence and metastatic rates in patients with advanced cancer, new therapeutic strategies must be developed to improve treatments. In some cases of oral cancers and tongue cancers, traditional treatments are difficult to implement, because the lesion is full of blood vessels, high risk of metastasis, and the patients may suffer from large surgical trauma and severe psychological stress. Consequently, exploring the molecular mechanisms at play is crucially important.

Moreover, we have rewritten the “discussion and conclusions” section and mentioned the potential strategies for TGF-β signaling inhibition. The corresponding paragraph was included as follows: In the tumor microenvironment, TGF-β is becoming an important and clinically actionable means of immune evasion. Meanwhile, TGF-β signaling inhibition is also an emerging strategy for cancer therapy. The three potential strategies for TGF-β signaling inhibition are:1) inhibiting TGF-β signaling components or modulating factors; 2) blocking the interaction between TGF-β and other signaling pathways in cancer; 3) normalizing tumor microenvironment homeostasis by down-regulating stromal stimulation resulting from excess TGF-β produced by tumor and tumor-related tissues.

Point 2: The review does not discuss any contrary evidence or studies that challenge the proposed roles of TGF-β. The inclusion of counter-arguments or contrasting viewpoints can provide a more balanced pers pective.

Response 2: We really appreciate your insightful comment. It is critical to discuss the contrary evidence, especially when the TGF-β signaling pathway molecules were targeted as therapeutic biomarkers. Therefore, we have added the counter-arguments and challenges for the TGF-β pathway targeted therapeutics as follows: Despite the accumulated evidence of TGF-β targeting therapeutics, it should also be noticed that inhibitors of TGF-β receptor kinases sometimes have poor pharmacokinetics and pharmacodynamics. They target TGF-β receptors non-specifically, and inhibit related type I receptors for several other TGF-β-related proteins equally effective nodal, activin, and myostatin [115], and may also inhibit other kinases, such as p38 MAPK [119]. The lack of specificity and poor pharmacokinetics of current TGF-β receptor kinase inhibitors lead to potential challenges with dosing strategies. Due to the aforementioned issues of the TGF-β pathway targeted therapeutics, it is critical to identify reliable biomarkers that predict response to TGF-β inhibitors.

In addition, we have revised the “discussion and conclusions” section about the limitation of the proposed roles of TGF-β inhibitors and the challenge of the TGF-β blocking progress in clinical trials. The revised paragraph is as follow: TGF-β pathway inhibitors have been investigated in the preclinical setting, some of which are now in the clinical development phase [127]. However, unlike preclinical results, TGF-β blocking progress in clinical trials has always been difficult, and many trials have failed to replicate success in animal models. Due to the dual role of the TGF-β signaling pathway, the window of effective TGF-β targeting is therefore evidently small, which poses a clear challenge in selecting patients at the right time. The ideal evaluation method will enable the identification of individuals who will benefit from TGF-targeted therapy, as well as excluding patients that TGF-targeting will create limited or even detrimental effects. Alternatively, another potential reason for the relative lack of success in clinical trials is that immune suppressive TGF-β signaling may be more nuanced than previously realized, and additional factors must be considered when designing anti-TGF-β therapies. Since the effects of TGF-β signaling are both localized and rapid [15], it is important to choose the dose and frequency of administration to support the sustained inhibition of the TGF-β receptors and prevent the reactivation of TGF-β signals in the tumor microenvironment. In addition, the identification of reliable and predictable biomarkers of response to TGF-β inhibitors is also a critical issue.

Point 3: Although the review discusses the potential of TGF-β as a therapeutic target, it does not provide a clear conclusion or suggest future research directions.

Response 3: We sincerely appreciate your comment. We have revised the “discussion and conclusion” section of the manuscript and added a future implication section at the end to provide a clear conclusion and future research direction. The revised conclusion is as follows: The TGF-β signaling plays a key role in cancer progression and high TGF-β signaling activity has been linked to resistance to multiple anticancer therapies, including chemotherapy and molecularly targeted therapies [127]. Enhanced intratumoral TGF-β signaling is also a barrier in patients in response to immunotherapy. Given the role of TGF-β signaling in non-pathophysiological processes, it is not surprising that systemic inhibition of TGF-β signaling would accompany the onset of adverse events. However, current early clinical trials of TGF-β inhibitors show that adverse events are generally controllable, such that therapies respond significantly and persistently when combined with immune checkpoint inhibitors. In the future, inhibition of TGF-β with dual-specific drugs or using drug interference with the composition of TGF-β signaling with restricted tissue expression are expected to lead to new therapies. Furthermore, combining TGF-β inhibitors with cytotoxic drugs, radiotherapy, adoptive T cell transfer therapy (such as CAR-T), immune checkpoint inhibitors (such as PD-1/PD-L1 antibodies), cancer vaccines, viral vectors, nanoparticles, or oncolytic viruses may provide additional opportunities.

And the revised future implications are as follows: Our understanding of oral cancer has been greatly improved by the developments in molecular biotechnology and human genetics. TGF-β signaling plays a complex role in oral cancer progression and development, as it can serve as a tumor promoter or suppressor, depending on the progression stage of oral cancer. However, much remains to be investigated in the future, particularly the cross-talk between the TGF-β pathway and other signaling pathways. Further elucidation of regulatory tumor-host interactions may identify more useful therapeutic targets and aid in the accurate timing of TGF-β-targeted therapy and clinical development of new TGF-β inhibitors.

TGF-β and other components of the TGF-β signaling pathway are also therapeutic antitumor candidates. Future work on the TGF-β signaling mechanisms that operate in oral cancer may reveal new therapeutic targets such as small-molecule TGF-β inhibitors, broaden the range of drug development and even prevent or cure oral cancer.

Point4: The review does not critically assess the quality or limitations of the cited studies, which is crucial for a more nuanced understanding of the topic.

Response4: Thanks for your comment. We downloaded the original research articles, confirmed that the experiment design and data were logical (please see the selection process in Materials and Methods section), and carefully quoted its conclusion. The comments in this article are more descriptive results in order to be rigorous. Moreover, we have rephrased the “TGF-β as a therapeutic biomarker” section according your suggestion. The corresponding changes to address this comment as follow: Despite the accumulated evidence of TGF-β targeting therapeutics, it should also be noticed that inhibitors of TGF-β receptor kinases sometimes have poor pharmacokinetics and pharmacodynamics. They target TGF-β receptors non-specifically, and inhibit related type I receptors for several other TGF-β-related proteins equally effective [115], and may also inhibit other kinases, such as p38 MAPK [119]. The lack of specificity and poor pharmacokinetics of current TGF-β receptor kinase inhibitors lead to potential challenges with dosing strategies.

Point 5: Given the dual role of TGF-β in tumor suppression and promotion, there could be a more balanced discussion on this aspect, including potential pitfalls and challenges in targeting TGF-β as a therapeutic strategy.

Response 5: Thank you very much for your comment. To address this comment, we have revised the manuscript in the “Discussion and Conclusions” section and discussed in detail the promising strategies for TGF-β signaling inhibition-based cancer treatment. We also mentioned the failure or challenges in the clinical trials of these drugs, trying to discuss the possible cause and influencing factors. The discussion is as follows: In the tumor microenvironment, TGF-β is becoming an important and clinically actionable means of immune evasion [126]. Meanwhile, TGF-β signaling inhibition is also an emerging strategy for cancer therapy. The three potential strategies for TGF-β signaling inhibition are:1) inhibiting TGF-β signaling components or modulating factors; 2) blocking the interaction between TGF-β and other signaling pathways in cancer; 3) normalizing tumor microenvironment homeostasis by down-regulating stromal stimulation resulting from excess TGF-β produced by tumor and tumor-related tissues.

TGF-β pathway inhibitors have been investigated in the preclinical setting, some of which are now in the clinical development phase [127]. However, unlike preclinical results, TGF-β blocking progress in clinical trials has always been difficult, and many trials have failed to replicate success in animal models. Due to the dual role of the TGF-β signaling pathway, the window of effective TGF-β targeting is therefore evidently small, which poses a clear challenge in selecting patients at the right time. The ideal evaluation method will enable the identification of individuals who will benefit from TGF-targeted therapy, as well as excluding patients that TGF-targeting will create limited or even detrimental effects. Alternatively, another potential reason for the relative lack of success in clinical trials is that immune suppressive TGF-β signaling may be more nuanced than previously realized, and additional factors must be considered when designing anti-TGF-β therapies. Since the effects of TGF-β signaling are both localized and rapid [15], it is important to choose the dose and frequency of administration to support the sustained inhibition of the TGF-β receptors and prevent the reactivation of TGF-β signals in the tumor microenvironment. In addition, the identification of reliable and predictable biomarkers of response to TGF-β inhibitors is also a critical issue.

The TGF-β signaling plays a key role in cancer progression and high TGF-β signaling activity has been linked to resistance to multiple anticancer therapies, including chemotherapy and molecularly targeted therapies [127]. Enhanced intratumoral TGF-β signaling is also a barrier in patients in response to immunotherapy. Given the role of TGF-β signaling in non-pathophysiological processes, it is not surprising that systemic inhibition of TGF-β signaling would accompany the onset of adverse events. However, current early clinical trials of TGF-β inhibitors show that adverse events are generally controllable, such that therapies respond significantly and persistently when combined with immune checkpoint inhibitors.

Round 2

Reviewer 3 Report

The authors have addressed most of my concerns.

Instances of colloquial stamenents.